# The impact of frailty and illness perceptions on quality of life among people living with HIV in Greece: A network analysis

**Anargyros Kapetanakis**[1]*, **Georgios Karakatsoulis**[1,2], **Dimitrios Kyrou**[1], **Iliana Ntourou**[1], **Nikolaos Vrontaras**[1], **Olga Tsachouridou**[3], **Maria Meliou**[4], **Dimitrios Basoulis**[5], **Konstantinos Protopapas**[6], **Vasilis Petrakis**[7], **Leonidia Leonidou**[8], **Ioannis Katsarolis**[9], **Simeon Metallidis**[3], **Maria Chini**[4], **Mina Psichogiou**[5], **Anastasia Antoniadou**[6], **Periklis Panagopoulos**[7], **Charalambos Gogos**[8], **Christina Karamanidou**[1]

1 Center for Research & Technology, Hellas, INAB, Thermi, Thessaloniki, Greece, 2 Department of Mathematics, University of Ioannina, Ioannina, Greece, 3 1st Department of Internal Medicine, AHEPA University Hospital, Medical School, Aristotle University of Thessaloniki, Thessaloniki, Greece, 4 3rd Department of Internal Medicine-Infectious Diseases Unit, "Korgialeneio-Benakeio" Red Cross General Hospital, Athens, Greece, 5 1st Department of Medicine, Laikon General Hospital, Medical School, National and Kapodistrian University of Athens, Athens, Greece, 6 4th Department of Medicine, Attikon General Hospital, Medical School, National and Kapodistrian University of Athens, Athens, Greece, 7 Department of Internal Medicine, University General Hospital, Democritus University of Thrace, Alexandroupolis, Greece, 8 Department of Internal Medicine and Infectious Diseases, University Hospital of Patras, Rio, Greece, 9 Medical Affairs, Gilead Sciences Hellas and Cyprus, Elliniko, Greece

* argi.kapten@certh.gr

**Data Availability Statement:** In compliance with GDPR mandates, our dataset, which encompasses patient data classified as sensitive, is subject to

## Abstract

### Objective

Despite the significant advances in healthcare, people living with HIV still face challenges that affect their quality of life (QoL), both in terms of their physical state as represented by frailty and of their illness perceptions (IP). The aim of this study was to unravel the associations between these constructs (QoL, frailty, IP).

### Methods

This multicenter, cross-sectional study included 477 people living with HIV (93% male; median age = 43 years, IQR = 51.7) from six HIV clinics in Greece. Frailty phenotype, QoL and IP were assessed using Fried's criteria, EuroQoL (EQ-5D-5L) and Brief Illness Perception Questionnaire (BIPQ), respectively. Network analysis model was utilized.

### Results

Among frailty criteria, exhaustion had the highest expected influence, while the strongest correlation concerns exhaustion and weak grip strength (pr = 0.14). Regarding the QoL items, usual activities displayed the highest expected influence. The correlations of pain/discomfort with mobility (pr = 0.31), and usual activities with self-care (pr = 0.34) were the strongest. For the BIPQ items, the strongest correlation was found between illness concern and emotional response (pr = 0.45), whereas the latter item was the one that displayed the

stringent access controls as determined by our institutional ethics committee and our ISO27001 certified ISMS. Access to this dataset is conditional upon obtaining explicit consent from the designated PI (ckaramanidou@gmail.com) as well as authorization from our institutional ethics committee (inab@certh.gr), thereby ensuring access is restricted solely to verified researchers in accordance with GDPR stipulations.

**Funding:** This study is a collaborative research project that is supported and funded by Gilead Sciences Hellas (Medical Affairs). There was no additional external funding received for this study.

**Competing interests:** I have read the journal's policy and the authors of this manuscript have the following competing interests: IK is an employee of Gilead Sciences Hellas and Cyprus (Medical Affairs). CK has received a grant from Gilead Sciences Hellas, paid to her institution, to support this collaborative study. IK contributed to the study design, the draft and the review of the manuscript, while no contribution was provided regarding data collection, data entry, data analysis and interpretation of findings. Additionally, this commercial affiliation does not alter our adherence to PLOS ONE policies on sharing data and materials. The specific role of this author is articulated in the 'author contributions' section

highest expected influence. Three communities were formed: 1) personal control, treatment control and coherence, 2) the frailty items with mobility, self-care, usual activities, and pain/discomfort, and 3) the rest BIPQ items with anxiety/depression. Identity displayed the highest bridge strength, followed by pain/discomfort, usual activities and consequences.

## Conclusions

The interplay between QoL, frailty, and IP in people living with HIV requires clinical attention. Self-reported exhaustion, slow walking speed, and low physical activity affect the physical QoL dimensions, while anxiety/depression is strongly associated with illness-related concern and perceived emotional effects, leading to psychological distress. Symptom management can improve QoL, and information on the disease and treatment can enhance control over the disease. Developing interventions to address QoL, frailty, and IP is crucial.

## Introduction

Despite the scientific advances that have turned human immunodeficiency virus (HIV) infection from a life-threatening disease to a chronic manageable condition [1] and the "undetectable equals untransmittable" life-transforming and empowering message [2], people living with HIV are still confronting challenges that can affect their quality of life (QoL) and health outcomes.

QoL refers to the impact of one's health on their perceived daily functioning and ability to live a fulfilling life [3]. QoL is a multi-domain construct as it is indicated by the numerous measurement tools and their varying facets (e.g., physical, psychological and social domains). People living with HIV experience lower QoL compared to the general population [4,5]. Ethnicity, homelessness, unemployment, consumption of alcohol and illicit drugs, hopelessness, negative self-image, sexual dissatisfaction, presence of physical symptoms and comorbidities, lack of social support, discrimination, stigma and poor adherence to therapy have been identified as risk factors that adversely affect QoL in recent studies [6–8]. In a large cross-sectional study (n = 3258) using one of the most prevalent QoL tools in clinical research, EuroQoL 5D [9], anxiety/depression was the most affected QoL domain in people living with HIV [4]. Moreover, a recent study showed that the risk of having a poor QoL with respect to physical functioning, bodily pain and general health was comparable with that of diabetes, but lower than in people living with rheumatoid arthritis. However, the odds of having poor mental health were higher in people living with HIV compared to people with either of these chronic illnesses [10].

People living with HIV constitute an ageing population experiencing age-related comorbidities and geriatric syndromes, such as frailty [11]. Frailty is a state in which there is low homeostatic capacity and increased vulnerability to stressors [12,13]. People living with HIV are more susceptible to frailty and are affected prematurely by it compared to the general population [14]. Furthermore, older age, multimorbidity, polypharmacy, diagnosis of acquired immunodeficiency syndrome (AIDS), and low current CD4+ cell count are predictors of frailty [11,15,16]. Therefore, frailty has been characterized as an indicator of physical weakness, "biological ageing", comorbidity burden and current immunological capacity in people living with HIV [12]. Apart from its great impact on QoL [17,18], frailty has been associated with negative health outcomes, including falls, fractures, disability, hospitalizations and mortality [19–23].

Besides the physical factors that were found to influence the QoL of people living with HIV, it is worth examining the impact of illness perceptions (IP), a psychological variable. IP is a construct which refers to how people perceive their disease (cognitive representation) and react to it emotionally (emotional representation), influencing the ways they cope, but also their adherence to therapy and, to an extent, their illness outcomes [24]. Additionally, it has been found to be associated with QoL [25–27]. As an example, people's perceptions of HIV having severe consequences and related symptoms have been linked with dysfunctional coping [28], while feeling in control of the disease and perceiving that HIV has mild consequences have been associated with healthy coping mechanisms and a positive psychological adjustment [29]. Negative IP of HIV, for example perceptions that one does not have personal control over the situation, were associated with experienced symptoms [30] and were negatively related to QoL [31]. Moreover, negative IP have been found to predict psychological distress, anxiety and depression in people living with cancer [32,33], which in turn can result in a lower QoL [25].

QoL, frailty and IP are multi-dimensional and encompass different sub-concepts. Much still remains to be explored regarding the nature of their inter-relationships. Thus, the aim of the current study is to unravel the associations of these constructs for people living with HIV in Greece by applying network analysis. This analysis will provide insights into which variables are directly related to each other, after partialling out all other variables. Moreover, it will be used in order to explore which variables are central/peripheral, and to identify clusters and patterns among the data. Deepening our understanding of these variables and their interconnections could facilitate the identification of areas with significant clinical value and potential intervention points for improvement of clinical outcomes and QoL for people living with HIV.

## Materials and methods

### Participants and procedures

This study is a secondary analysis of data collected in the "HIV Holistic Assessment" program in Greece, which was the first ever attempt to map frailty among people living with HIV at a national scale [11]. This was a multicenter, cross-sectional study conducted in six HIV clinics in major cities in Greece (Athens, Thessaloniki, Alexandroupoli, Patras). Non-probability, consecutive sampling was employed and data collection was performed for the period from September 2019 to September 2020. Participants were adult people living with HIV who are outpatients in the participating HIV clinics. There were no exclusion criteria for participation.

Participants were asked to be enrolled to the study by their regular HIV clinician during their routine appointments. Following their routine appointment, their clinician performed a physical assessment for frailty and gave participants questionnaires to fulfill, which were supplied to the research team along with demographic and clinical data from the participants' medical records. All data were collected and fully anonymized by clinicians, who were the only ones from the authors with access to full data. The research team conducted the current study analyzing the anonymized collected data for the period from October 2022 to May 2023.

### Measurements

Frailty was assessed using the Fried Frailty Phenotype (FFP) as proposed by Fried et al [13]. FFP utilizes five criteria: weakness in grip strength, slowness in gait speed, low level of physical activity, self-reported exhaustion, and unintentional weight loss. Based on the fulfilled criteria, patients are classified as robust (zero), pre-frail (one or two) and frail (three or more). Physical activity levels were calculated using the Greek version of the International Physical Activity

Questionnaire (IPAQ-Gr) [34]. A brief description of the used frailty criteria is provided in S1 Appendix.

Quality of life was assessed using the Greek version of EuroQoL (EQ-5D-5L) [35]. The questionnaire consists of five dimensions of health, including mobility, self-care, usual activities, pain/discomfort, and anxiety/depression. All items are rated using a 5-point Likert scale (one to five) whereby higher scores indicate a lower Quality of Life.

Illness perceptions were assessed using the Greek version of Brief Illness Perception Questionnaire (BIPQ) [36]. The questionnaire consists of nine items. Five items evaluate cognitive illness representation: consequences–item 1 (how much illness affects patient's life), timeline–item 2 (how long illness will continue), personal control–item 3 (how much control patient has over their illness), treatment control–item 4 (how much treatment can help patient's illness) and identity–item 5 (how much patient experiences symptoms from their illness). Two items evaluate emotional representation: illness concern–item 6 (how concerned patient is about their illness) and emotional response–item 8 (how much illness affects patient emotionally). One item evaluates illness comprehensibility: coherence–item 7 (how well patient understands their illness). Lastly, item 9 is an open question regarding the three most important factors that caused the patient's illness. Apart from the open question, all items are rated using a 11-point Likert scale (zero to ten). Higher scores indicate more negative illness perceptions. Item 9 was not used in our analysis as it includes qualitative data.

## Ethics

Prior to enrollment, participants provided an informed written consent for study participation and access to data from their medical records after being informed about the nature and the purpose of the study, the protection of their confidentiality and anonymity, and their right to remove themselves from the study. The study was performed in agreement with the Declaration of Helsinki and the General Data Protection Regulation (GDPR), as stipulated in EU Regulation 2016/679. Ethical approval was granted by the Research Ethics Committee of the Institute of Applied Biosciences at the Center for Research and Technology, Hellas, as well as by the Research Ethics Committee of each participating clinical site.

## Statistical analysis

In descriptive statistics, means (M) and standard deviations (SD) were used for the numeric variables whereas frequencies and relative frequencies were used for the categorical ones.

The associations of BIPQ and QoL items (ordinal variables) with the frailty criteria (binary variables) were tested using a Kruskal-Wallis test. Spearman coefficient was used to estimate correlations between (and within) BIPQ and QoL items. The significance level was set to 5%, and false discovery rate (FDR) correction was used in case of multiple comparisons.

Concerning the network analysis, in each network the nodes represent the variables under investigation (frailty, QoL, BIPQ items), whereas the edges were calculated through the partial correlation coefficients, in order to distinguish the direct from the indirect associations and investigate potential mediation effects. The width of an edge is analogous to the magnitude of the association, the edge color represents its direction (blue for positive, red for negative), and the node color represents the community in which it participates. Network regularization was conducted through graphical Lasso algorithm, with the tuning parameter being selected as the one that minimizes the extended Bayesian Information Criterion (EBIC).

The role and importance of each variable was investigated via three centrality measures (closeness, betweenness and expected influence), which were calculated using the z-scores.

Community detection was performed through the fast-greedy algorithm, and bridge strength was used to examine the variables that act as bridges connecting distinct communities.

All statistical analyses were conducted in R (version 4.1.3). Any missing data were omitted. Network estimation was performed using bootnet [37] and qgraph [38] packages, igraph [39] was used to derive communities and network tools [40] for calculating bridge centrality measures. Data visualization was performed through qgraph and ggplot2 [41].

## Results

### Descriptive statistics

**Demographics and clinical information.** Our sample included 477 participants (444 males, 93%). The mean age was 43.9±11.4 years and 93.5% had undetectable HIV viral load. Detailed information about demographic and clinical data is available in the Table 1.

**Frailty.** Among frailty criteria, weak grip strength was the most prevalent (26.8%), followed by self-reported exhaustion (9.7%), low physical activity (8.7%), slow walking speed (4.4%) and unintentional weight loss (2.9%) (Table 2). Based on the above criteria, 285/459 (62.1%) participants were robust, 155/459 (33.8%) pre-frail and 19/459 (4.1%) frail.

**Quality of life.** Table 3 shows the results of QoL assessment. Anxiety/depression was the most negatively affected domain, as it was the most frequently reported with minor problems and/or more severe ones, followed by pain/discomfort. This was followed by mobility and usual activities, while self-care was the least impacted domain, with 97.7% of participants reporting no problems.

### Illness perceptions

Table 4 shows descriptive statistics for the BIPQ items and the overall score. The items that indicated the most negative perceptions were timeline (M = 8.48, SD = 2.43), which is expected due to HIV's chronic and incurable nature, followed by illness concern (M = 5.22, SD = 3.18), emotional response (M = 4.81, SD = 3.26), and consequences (M = 4.26, SD = 3.08). On the other hand, participants mainly reported that treatment can help with their illness (treatment control, M = 0.76, SD = 1.46) and that they understand their illness (coherence, M = 1.73, SD = 2.13).

### Correlations between measures

In this section, we aim to investigate the associations between QoL dimensions, frailty criteria and BIPQ items. To this end, we conducted all pair-wise comparisons between these constructs and are presented below.

**Frailty and QoL.** Regarding the association between QoL and frailty criteria, the analysis showed that anxiety/depression is related only to self-reported exhaustion. Specifically, people who feel exhausted display higher levels of anxiety/depression. As for the rest QoL dimensions, they are associated with all frailty criteria, except for weak grip strength. Exception constitutes the non-significant correlation between pain/discomfort and low physical activity. The above correlations are shown in Table 5.

**Frailty and BIPQ.** Concerning the correlations between frailty criteria and BIPQ items, self-reported exhaustion was associated with consequences, emotional response, illness concern and identity. Unintentional weight loss was associated with identity and treatment control. Slow walking speed was associated with identity. Lastly, low physical activity was associated with consequences. Notably, no significant correlations were found between weak grip strength and BIPQ items. The results are displayed in Table 6.

**Table 1. Demographic and CLINICAL information.**

| Variable | Category | Mean | SD |
|---|---|---|---|
| Age | Years | 43.9 | 11.4 |
| Weight | Kg | 80.8 | 15.2 |
| BMI | Kg/m$^2$ | 25.8 | 4.5 |
|  |  | **N** | **%** |
| Sex | Female | 33 | 7 |
|  | Male | 444 | 93 |
| Education level | Primary | 57 | 12 |
|  | Secondary | 149 | 31.4 |
|  | Post-secondary | 118 | 24.8 |
|  | Tertiary | 151 | 31.8 |
| Being married | Yes | 76 | 16.0 |
| Having children | Yes | 70 | 14.7 |
| Transmission group | Sex between men and women | 80 | 16.8 |
|  | People who inject drugs | 15 | 3.2 |
|  | Sex between men | 370 | 77.7 |
|  | Other | 11 | 2.3 |
| Recreational Drugs | Yes | 63 | 13 |
| Chemsex | Yes | 27 | 6 |
| Current Employment | Yes | 301 | 63 |
| Smocking | Yes | 218 | 46 |
| Alcohol | Yes | 46 | 9.9 |
| History of AIDS | Yes | 86 | 18.1 |
| CD4 cell count at last measurement | <200 cells/μL | 17 | 3.6 |
| CD4 cell count at diagnosis | <200 cells/μL | 108 | 23.1 |
| Undetectable HIV viral load | <50 copies/mL | 429 | 93.5 |
| Opportunistic infections | Yes | 31 | 6.5 |
| Sexually transmitted infections | Yes | 191 | 40 |
| Musculoskeletal disorders | Yes | 98 | 20.6 |
| Endocrine/ Metabolic disorders | Yes | 159 | 33.3 |
| Circulatory disorders | Yes | 62 | 13 |
| Genitourinary disorders | Yes | 36 | 7.6 |
| Liver disorders | Yes | 48 | 10.1 |
| Neurological disorders | Yes | 15 | 3.2 |
| Mental disorders | Yes | 58 | 12.2 |
| Cancer | Yes | 36 | 7.6 |
| Non-HIV medications | 0–1 | 368 | 77.1 |
|  | 2–3 | 70 | 14.7 |
|  | 4+ | 39 | 8.2 |

**Table 2. Frailty criteria assessment.**

| Frailty criterion | n | Non-frail n (%) | Frail n (%) |
|---|---|---|---|
| Exhaustion | 476 | 430 (90.3%) | 46 (9.7%) |
| Low physical activity | 461 | 421 (91.3%) | 40 (8.7%) |
| Slow walking | 475 | 454 (95.6%) | 21 (4.4%) |
| Weak Grip Strength | 477 | 349 (73.2%) | 128 (26.8%) |
| Weight loss | 476 | 462 (97.1%) | 14 (2.9%) |

Table 3. Quality of life.

| | Anxiety/ Depression n (%) | Mobility n (%) | Pain/ Discomfort n (%) | Self-Care n (%) | Usual Activities n (%) |
|---|---|---|---|---|---|
| Level 1 (No problems) | 122 (25.9) | 385 (81.4) | 302 (64.1) | 462 (97.7) | 405 (86) |
| Level 2 (Slight problems) | 151 (32.1) | 66 (14) | 115 (24.4) | 8 (1.7) | 54 (11.5) |
| Level 3 (Moderate problems) | 142 (30.1) | 11 (2.3) | 39 (8.3) | 1 (0.2) | 8 (1.7) |
| Level 4 (Severe problems) | 40 (8.5) | 11 (2.3) | 14 (3) | 2 (0.4) | 4 (0.8) |
| Level 5 (Extreme problems/ unable to do) | 16 (3.4) | 0 (0) | 1 (0.2) | 0 (0) | 0 (0) |
| Total | 471 (100) | 473 (100) | 471 (100) | 473 (100) | 471 (100) |

**QoL and BIPQ (between and within).** Turning now to the correlations between and within QoL and BIPQ items, the results indicated that the majority were statistically significant (ranging from small to high correlations). Specifically, within QoL items the highest correlations were between mobility and usual activities (r = 0.45), self-care and usual activities (r = 0.4), and mobility and pain/discomfort (r = 0.38). Within BIPQ items, the highest correlations were between illness concern and emotional response (r = 0.71), consequences and

Table 4. Illness perceptions.

| Item | n | Mean | SD |
|---|---|---|---|
| Coherence | 473 | 1.73 | 2.13 |
| Consequences | 473 | 4.26 | 3.08 |
| Emotional Response | 473 | 4.81 | 3.26 |
| Illness Concern | 473 | 5.22 | 3.18 |
| Identity | 473 | 1.98 | 2.54 |
| Personal Control | 473 | 2.37 | 2.51 |
| Treatment Control | 473 | 0.76 | 1.46 |
| Timeline | 473 | 8.48 | 2.43 |
| Overall score | 473 | 29.6 | 12.02 |

Table 5. Associations between frailty and quality of life.

| | | Anxiety/Depression | | | Mobility | | | Pain/Discomfort | | | Self-Care | | | Usual Activities | | |
|---|---|---|---|---|---|---|---|---|---|---|---|---|---|---|---|---|
| | | Mean | SD | p-value* | Mean | SD | p-value | Mean | SD | p-value | Mean | SD | p-value | Mean | SD | p-value |
| **Weight Loss** | **Non-frail** | 2.33 | 1.05 | 0.33 | 1.24 | 0.59 | **0.02** | 1.49 | **0.77** | **<0.01** | 1.03 | **0.24** | **<0.01** | 1.16 | 0.47 | **<0.01** |
| | **Frail** | 2 | 1 | | 1.85 | 0.99 | | 2.31 | 1.03 | | 1.23 | 0.44 | | 1.61 | 0.65 | |
| **Slow Walking** | **Non-frail** | 2.31 | 1.05 | 0.454 | 1.2 | 0.5 | **<0.001** | 1.45 | **0.71** | **<0.001** | 1.02 | **0.15** | **<0.001** | 1.15 | 0.43 | **0.009** |
| | **Frail** | 2.5 | 1.1 | | 2.45 | 1.32 | | 2.7 | 1.22 | | 1.4 | 0.94 | | 1.6 | 1 | |
| **Weak Grip Strength** | **Non-frail** | 2.27 | 1.01 | 0.39 | 1.21 | 0.51 | 0.36 | 1.45 | 0.71 | 0.2 | 1.02 | 0.15 | 0.2 | 1.14 | 0.42 | 0.15 |
| | **Frail** | 2.44 | 1.16 | | 1.38 | 0.83 | | 1.68 | 0.96 | | 1.08 | 0.41 | | 1.26 | 0.6 | |
| **Low Physical Activity** | **Non-frail** | 2.31 | 1.05 | 0.441 | 1.21 | 0.55 | **<0.001** | 1.49 | 0.76 | 0.198 | 1.02 | **0.21** | **0.005** | 1.14 | 0.41 | **<0.001** |
| | **Frail** | 2.4 | 0.96 | | 1.73 | 1.04 | | 1.77 | 1.04 | | 1.15 | 0.53 | | 1.55 | 0.88 | |
| **Exhaustion** | **Non-frail** | 2.26 | 1.03 | **0.004** | 1.19 | **0.51** | **<0.001** | 1.42 | **0.69** | **<0.001** | 1.01 | **0.14** | **<0.001** | 1.12 | 0.37 | **<0.001** |
| | **Frail** | 2.83 | 1.19 | | 1.84 | 1.07 | | 2.37 | 1.09 | | 1.23 | 0.68 | | 1.65 | 0.9 | |

* Bold values represent all p-values <0.05 (The significant level was set to a = 5%).

**Table 6. Associations between frailty and illness perceptions.**

| | | COH Mean | COH SD | COH p-value* | CONS Mean | CONS SD | CONS p-value | ER Mean | ER SD | ER p-value | IC Mean | IC SD | IC p-value | ID Mean | ID SD | ID p-value | PC Mean | PC SD | PC p-value | TC Mean | TC SD | TC p-value | TIME Mean | TIME SD | TIME p-value | Overall Score Mean | Overall Score SD | Overall Score p-value |
|---|---|---|---|---|---|---|---|---|---|---|---|---|---|---|---|---|---|---|---|---|---|---|---|---|---|---|---|---|
| Weight Loss | Non-frail | 1.72 | 2.11 | 0.633 | 4.22 | 3.08 | 0.11 | 4.77 | 3.27 | 0.108 | 5.18 | 3.18 | 0.128 | 1.95 | 2.54 | **0.025** | 2.33 | 2.49 | 0.097 | 0.72 | 1.42 | **0.025** | 8.49 | 2.43 | 0.77 | 29.36 | 11.99 | 0.01 |
| | Frail | 2.23 | 2.71 | | 5.77 | 2.98 | | 6.46 | 2.79 | | 6.62 | 3.02 | | 3.15 | 2.12 | | 3.77 | 2.77 | | 2 | 2.42 | | 8.08 | 2.63 | | 38.08 | 11.09 | |
| Slow Walking | Non-frail | 1.73 | 2.11 | 0.392 | 4.2 | 3.04 | 0.075 | 4.77 | 3.24 | 0.392 | 5.16 | 3.15 | 0.09 | 1.89 | 2.47 | **0.009** | 2.32 | 2.47 | 0.392 | 0.78 | 1.49 | 0.081 | 8.47 | 2.44 | 0.495 | 29.3 | 11.92 | 0.022 |
| | Frail | 1.65 | 2.64 | | 5.9 | 3.73 | | 5.55 | 3.8 | | 6.5 | 3.71 | | 3.95 | 3.22 | | 3.1 | 3.02 | | 0.2 | 0.52 | | 8.7 | 2.27 | | 35.55 | 13.65 | |
| Weak Grip Strength | Non-frail | 1.74 | 2.1 | 0.531 | 4.27 | 4.23 | 0.87 | 4.7 | 3.22 | 0.386 | 5.17 | 3.1 | 0.618 | 1.92 | 2.53 | 0.44 | 2.28 | 2.45 | 0.386 | 0.81 | 1.53 | 0.358 | 8.46 | 2.44 | 0.86 | 29.35 | 11.9 | 0.348 |
| | Frail | 1.68 | 2.22 | | 3.07 | 3.12 | | 5.11 | 3.36 | | 5.36 | 3.4 | | 2.15 | 2.54 | | 2.62 | 2.66 | | 0.61 | 1.26 | | 8.55 | 2.4 | | 30.25 | 12.41 | |
| Low Physical Activity | Non-frail | 1.71 | 2.18 | 0.158 | 4.13 | 3.08 | **0.019** | 4.75 | 3.27 | 0.261 | 5.18 | 3.19 | 0.353 | 1.89 | 2.48 | 0.136 | 2.33 | 2.51 | 0.239 | 0.7 | 1.39 | 0.201 | 8.49 | 2.44 | 0.353 | 29.15 | 12.05 | 0.065 |
| | Frail | 1.93 | 1.57 | | 5.5 | 2.82 | | 5.42 | 3.12 | | 5.68 | 3.07 | | 2.55 | 2.63 | | 2.67 | 2.35 | | 0.98 | 1.44 | | 8.18 | 2.49 | | 32.9 | 10.8 | |
| Exhaustion | Non-frail | 1.75 | 2.15 | 0.314 | 4.04 | 3.01 | **<0.001** | 4.64 | 3.22 | **0.001** | 5.01 | 3.14 | **<0.001** | 1.85 | 2.48 | **0.001** | 2.3 | 2.48 | 0.097 | 0.74 | 1.47 | 0.663 | 8.44 | 2.45 | 0.318 | 28.76 | 11.79 | **<0.001** |
| | Frail | 1.42 | 1.92 | | 6.39 | 3 | | 6.54 | 3.21 | | 7.28 | 2.84 | | 3.23 | 2.82 | | 3.05 | 2.74 | | 0.88 | 1.43 | | 8.84 | 2.16 | | 37.63 | 11.54 | |

* Bold values represent all p-values <0.05 (The significant level was set to a = 5%).

Abbreviations: COH, Coherence; CONS, Consequences; ER, Emotional response; IC, Illness concern; ID, Identity; PC, Personal control; TC, Treatment control; TIME, Timeline.

**Table 7. Associations between and within quality of life and illness perceptions.**

| Variable | 1 | 2 | 3 | 4 | 5 | 6 | 7 | 8 | 9 | 10 | 11 | 12 |
|---|---|---|---|---|---|---|---|---|---|---|---|---|
| 1. Mobility | 1 | | | | | | | | | | | |
| 2. Self-Care | 0.27** | 1 | | | | | | | | | | |
| 3. Usual Activities | 0.45** | 0.4** | 1 | | | | | | | | | |
| 4. Pain/ Discomfort | 0.38** | 0.23** | 0.32** | 1 | | | | | | | | |
| 5. Anxiety/ Depression | 0.08 | 0 | 0.21** | 0.23** | 1 | | | | | | | |
| 6. Consequences | 0.17** | 0.16** | 0.27** | 0.27** | 0.29** | 1 | | | | | | |
| 7. Timeline | 0.07 | -0.03 | 0.02 | 0.11* | 0.14** | 0.12** | 1 | | | | | |
| 8. Identity | 0.29** | 0.17** | 0.37** | 0.33** | 0.19** | 0.37** | 0.11** | 1 | | | | |
| 9. Illness Concern | 0.11* | 0.14** | 0.17** | 0.14** | 0.29** | 0.59** | 0.18** | 0.35** | 1 | | | |
| 10. Emotional Response | 0.09 | 0.1* | 0.17** | 0.16** | 0.4** | 0.64** | 0.19** | 0.33** | 0.71** | 1 | | |
| 11. Personal control | 0.14** | 0.14** | 0.21** | 0.19** | 0.14** | 0.25** | 0.04 | 0.36** | 0.24** | 0.24** | 1 | |
| 12. Treatment control | 0.1* | 0.1* | 0.16** | 0.08 | 0.08 | 0.17** | -0.09 | 0.33** | 0.17** | 0.15** | 0.38** | 1 |
| 13. Coherence | 0 | 0.11* | 0.11* | 0.03 | 0.1* | 0.23** | -0.1 | 0.14** | 0.12** | 0.07 | 0.3** | 0.31** |

* p < 0.05

** p < 0.01.

emotional response (r = 0.64), and consequences and illness concern (r = 0.59). Between QoL and BIPQ items, the highest correlations were between identity and usual activities (r = 0.37), and identity and pain/discomfort (r = 0.33). The results are explicitly displayed in Table 7.

## Network analysiss

The results obtained through the univariate analysis led us to further investigate how the items of these constructs (QoL, frailty and BIPQ) interact with each other. To this end, we used network analysis in order to distinguish the direct from the indirect associations and investigate potential mediation effects. The resulting network is presented in Fig 1.

**Centrality analysis.** Concerning the frailty criteria, exhaustion is positively directly associated with weight loss, slow walking speed and low grip strength, being the criterion with the highest expected influence. Within the frailty criteria, the strongest correlation concerns the exhaustion and low grip strength (pr = 0.14). Regarding the QoL items, pain/discomfort and usual activities are positively directly associated with all the other items, whereas the latter also displays the highest expected influence. The correlations of pain/discomfort with mobility, and usual activities with self-care are the strongest (pr = 0.31 and pr = 0.34, respectively). As for the BIPQ items, two subnetworks seem to appear: one comprised of personal control, treatment control and coherence, and the other comprised of the rest of the items. Illness concern is directly correlated with almost all of the other BIPQ items (except for coherence). The strongest correlation is found between illness concern and emotional response (pr = 0.45), whereas the latter item is the one that displays the highest expected influence.

Apart from the intracorrelations observed, the network reveals significant correlations between items of different measures. Overall, emotional response and usual activities are the most central nodes in terms of expected influence, followed by consequences, illness concern,

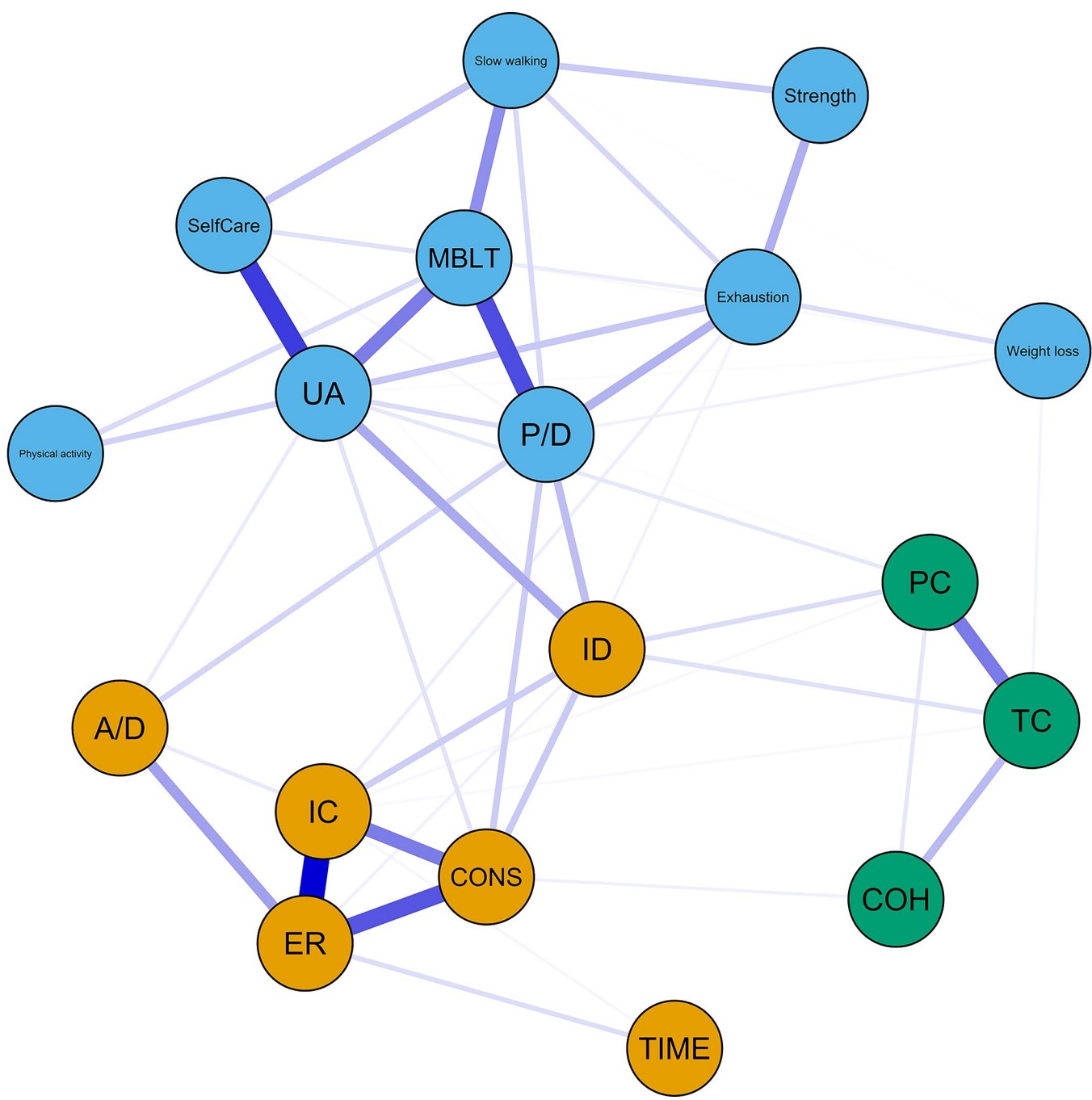

**Fig 1. Network of frailty, quality of life and illness perception.** Abbreviations: A/D, Anxiety/Depression; UA, Usual activities; P/D, Pain/Discomfort; MBLT, Mobility; COH, Coherence; CONS, Consequences; ER, Emotional response; IC, Illness concern; ID, Identity; PC, Personal control; TC, Treatment control; TIME, Timeline.

mobility and pain/discomfort. This result indicates that the aforementioned items are the most influential components in the network. In contrast, timeline, weight loss and physical activities display the lowest expected influence values, indicating that they have minor impact on the network. Concerning the betweenness centrality, identity and pain/discomfort display the highest values, indicating that they act as "bridges" connecting different nodes (Fig 2).

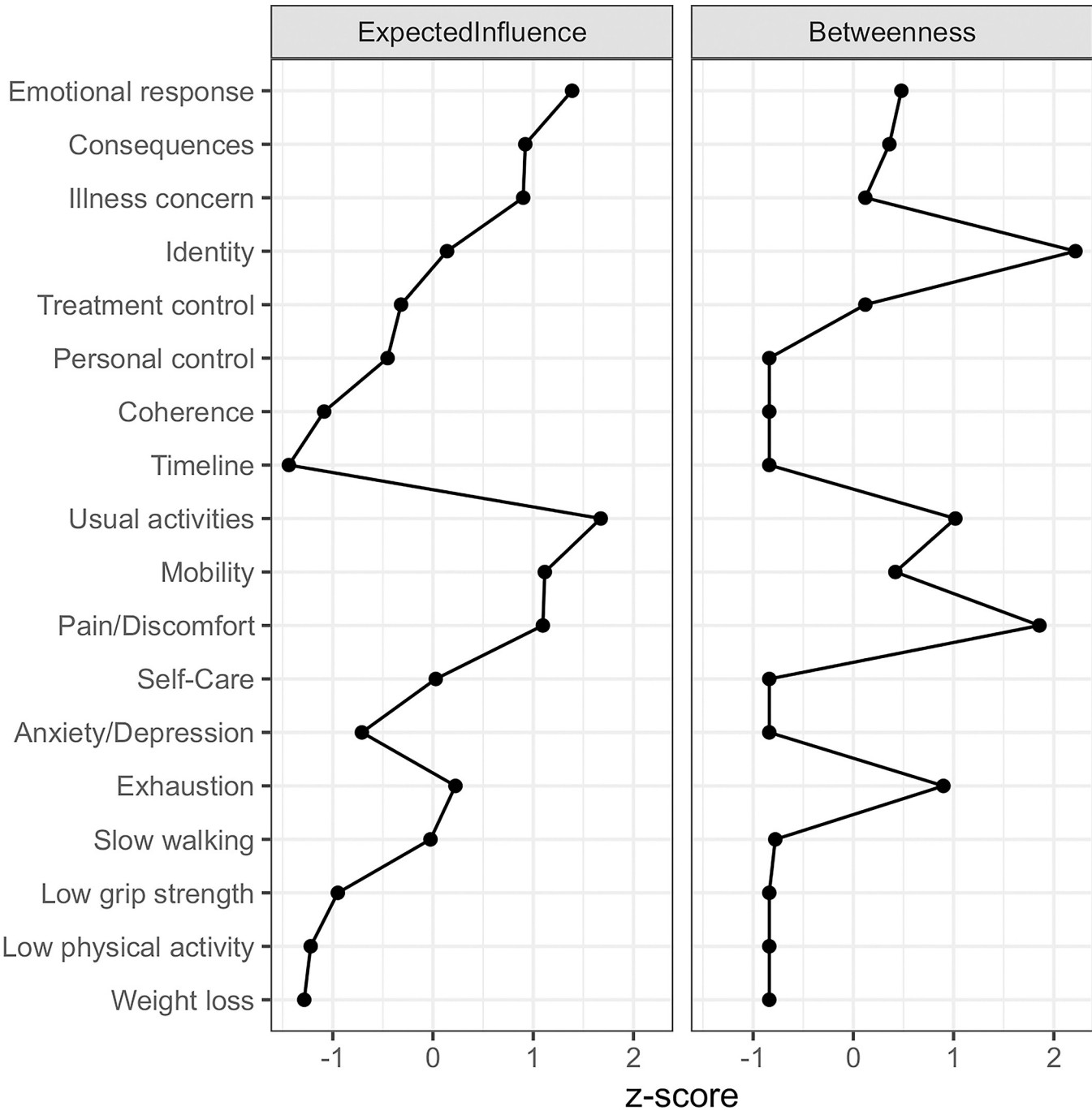

**Fig 2. Centrality measures.**

**Communities.** Community detection showed three different communities, with the one comprised by personal control, treatment control and coherence, a second comprised by the frailty items along with mobility, self-care, usual activities and pain/discomfort, and the last comprised of the rest BIPQ items along with anxiety/depression. Identity displays the highest bridge strength, followed by pain/discomfort, usual activities and consequences, a result that implies that these items comprise the connections between the different communities (Fig 3).

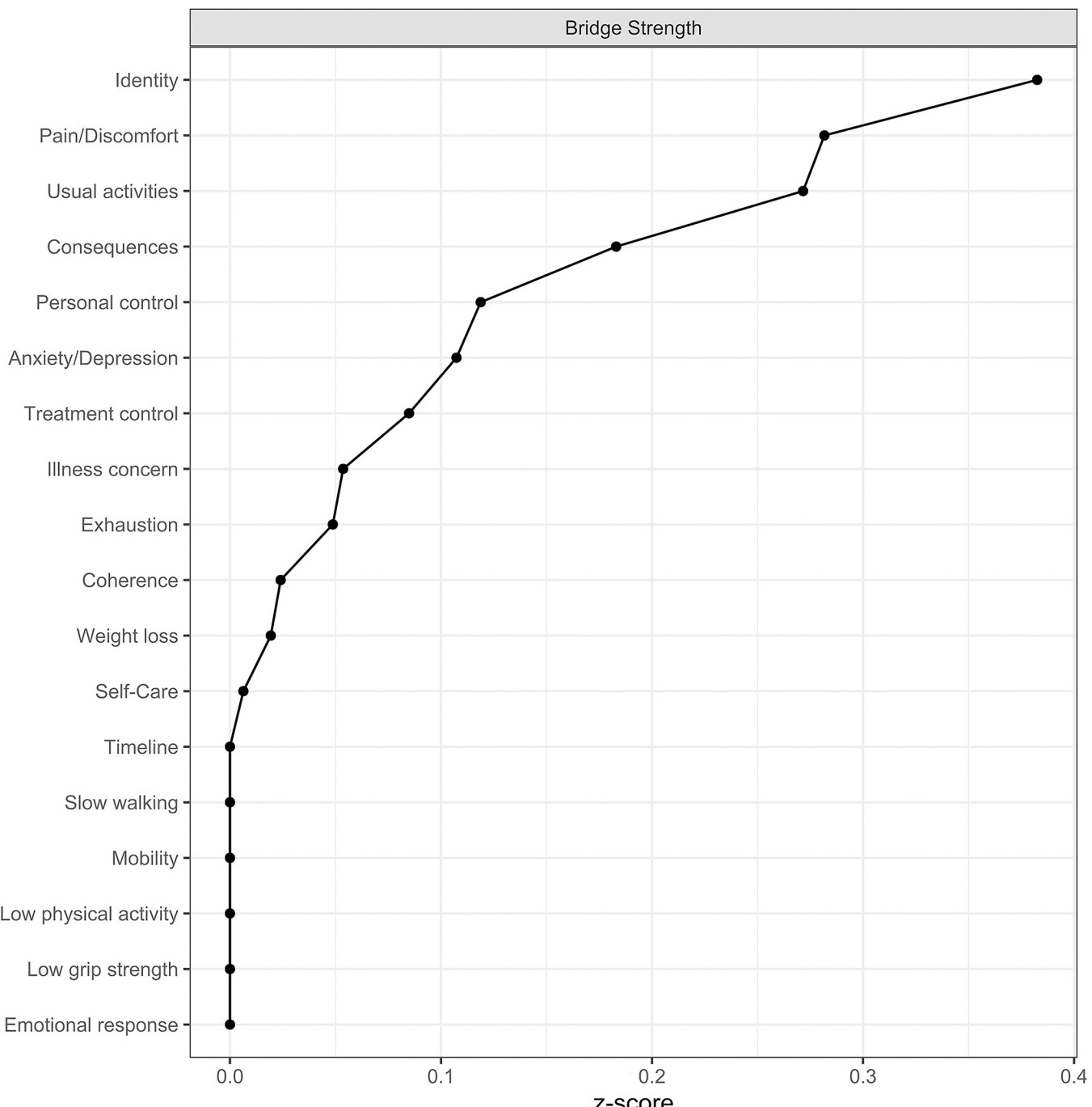

**Fig 3. Bridge strength.**

## Discussion

The current study examined the impact of frailty and IP on QoL of people living with HIV in Greece by investigating their relationships. Using traditional statistics, multiple associations between and within the three constructs were revealed. In order to have a clearer understanding of the overlapping relationships, networks analysis was utilized, a method which provides

a meticulous analysis of the dynamic relations between BIPQ, EuroQol and FFP. This allowed for the identification of central variables, clusters as well as points of high clinical value to be derived from a thorough analysis of their dynamic relations. To our knowledge, this is the first study that investigates these constructs using the Networks Theory in the scientific area of HIV.

Firstly, in the network (see Fig 1) four out of five QoL dimensions, namely self-care, usual activities, mobility, and pain/discomfort, play a central role with strong associations between them. Therefore, people living with HIV who are experiencing a decline in one of these domains are likely to face a negative impact on the other QoL domains as well. This pattern of HIV's impact on QoL requires clinical attention for early prevention, detection, and intervention. Previous studies have shown that self-care, usual activities, mobility, and pain/discomfort have lower scores in people living with HIV compared to the general population [4] and that pain/discomfort is associated with poorer outcomes [42] and increased risk of impairment in mobility, self-care and usual activities in people living with HIV [43]. Thus, providing care that addresses these domains might be a relevant target for interventions to improve the QoL of people living with HIV. For example, clinicians could screen for pain/discomfort or other issues that may negatively influence their usual activities and suggest appropriate measures.

Overall, the QoL dimensions of self-care, usual activities, mobility, and pain/discomfort form a cluster with all frailty criteria, designating a more physical and social-functioning component of the network. This finding confirms the impact of frailty on QoL (20,21) and shows that frailty seems to have a greater impact on physical QoL dimensions. Among frailty criteria, self-reported exhaustion, slow walking speed and low physical activity have a direct influence on QoL dimensions, affecting them notably. Therefore, these three criteria can be considered important points for intervention in order to improve frailty phenotype of people living with HIV and consequently their QoL [44–46].

Weakness in grip strength is the most prevalent frailty criterion in our study, which is consistent with the literature [47,48]. People living with HIV experience a higher decline in grip strength compared to the general population [49]. However, in the network analysis, we found that this variable has a less central position and has less influence on QoL than slow walking speed and self-reported exhaustion. Additionally, even with traditional statistics, no strong associations were found with either the items of the QoL questionnaire or with the BIPQ. This raises reasonable questions about the validity and specificity of this criterion in assessing frailty phenotype, but also challenges the reliability of the actual measurement of grip strength using a dynamometer. In both cases, it would be useful to examine another means of assessing strength decrease and muscle mass loss in people living with HIV.

The remaining QoL domain, anxiety/depression, was the most affected in the present study, in line with previous research [4]. Using traditional statistics, anxiety/depression was strongly associated with all BIPQ items, except for treatment control. The relationship between anxiety/depression with BIPQ has also been demonstrated by a recent meta-analysis [24]. Network analysis revealed that anxiety/depression comprised a cluster with five out of eight BIPQ items, namely emotional response, illness concern, consequences, identity, and timeline. However, anxiety/depression was directly related just to illness concern and emotional response: the emotional representation component of the BIPQ [36]. This observation renders illness concern and emotional response potential points for IP-based interventions. For instance, a multidisciplinary team of healthcare professionals could elicit and modify their dysfunctional beliefs, teach them techniques to manage anxiety and depression and help them in behaviour change required with regards to their lifestyle [50].

Network analysis revealed that the correlation between the perception of HIV symptoms (identity) and anxiety/depression (see Table 7) is explained by the influence of illness concern,

usual activities and pain/discomfort (see Fig 1). Similarly, usual activities and pain/discomfort explain the relationship between anxiety/depression and exhaustion (see Table 7 and Fig 1). Thus, there are indications that people living with HIV are psychologically affected when they worry about their illness, when they experience pain/discomfort or when their usual activities, such as working or doing leisure activities, are hampered. The presence of physical symptoms or exhaustion per se are not sufficient conditions on their own for anxiety/depression to develop.

Perception of symptoms (identity) was the variable that had the most mediating role among BIPQ items and the higher bridge strength between clusters in the network (see Figs 2 and 3). One possible explanation is that the way individuals perceive the symptoms of their illness plays an important role and is influenced or influences not only the rest of their illness perceptions, but also their QoL. In clinical practice, the identification and treatment of symptoms among people's living with HIV should be the focus of attention, as their relief could offer improvement in other domains as well.

Lastly, a third cluster was formed by coherence, personal control and treatment control. This cluster aligns with previous network and other psychometric analyses of the BIPQ on people living with HIV [51,52], which suggest that these items form a factor of "control" in the BIPQ. For people living with HIV, having a comprehensive understanding of their disease, acknowledging the effectiveness of treatment, and adhering to it are critical for achieving good control over their disease. This highlights the significance of acknowledging the value of U = U and its benefits not only at a personal level by improving health and well-being, but also at a societal level, as it has the potential to reduce HIV stigma, enhance HIV prevention efforts, and ultimately contribute to the collective aim of ending the HIV epidemic [53]. Interestingly, our study findings indicate that coherence, personal control and treatment control are better in people living with HIV than in other chronic diseases, such as diabetes and rheumatoid arthritis [54–56]. It seems that the emotional component of people's living with HIV IP is the most impacted, along with the anxiety/depression domain of their QoL.

## Limitations

This study has several limitations. The cross-sectional design of the study limits the interpretation of the found relationships, and a prospective longitudinal study is needed in order to assess the dynamic relations over time. The recruitment process was partially affected by the COVID-19 pandemic, possibly resulting in underrepresentation of the most vulnerable or frail patients, who may have opted to stay at home due to safety concerns (sampling bias). The COVID-19 pandemic may have influenced the QoL of our participants independently. By performing sub-network analysis including only the participants recruited before the first lockdown, we found similar results regarding the associations (S1 Fig) indicating that our findings were not affected by COVID-19 pandemic. Women constituted only the 6.9% of our sample, which is considerably lower than the global prevalence (53%), but closer to the respective prevalence in Greece (17%) [57] Injection drug use was the means of transmission for 3.2% of study participants, which is lower than the frequency of 10–15% in our patient population, based on our clinical experience. This population needs special attention as it is characterized by specific risk factors that determine their health status.

## Conclusions

The interplay between QoL, frailty and IP among individuals living with HIV warrants close attention from clinicians in routine clinical care beyond the achievement of viral suppression. Among frailty criteria, self-reported exhaustion, slow walking speed and low physical activity directly affect the physical QoL dimensions. Meanwhile, anxiety/depression is strongly

associated with concern about illness and perceived emotional effect. People living with HIV appear psychologically affected when they worry about their illness, have difficulties in doing usual activities and experience pain/discomfort. Furthermore, the identification and management of symptoms should be the focus of clinical attention, as their alleviation could improve QoL. Finally, providing information on the disease and treatment effectiveness can help people gain better control over their disease. Developing and implementing interventions to target QoL dimensions, as well as frailty and illness perceptions, are of paramount importance to empower people living with HIV to optimize their health status.

## Supporting information

**S1 Checklist. STROBE Statement—Checklist of items that should be included in reports of observational studies.**
(PDF)

**S1 Appendix. Description of FFP criteria.**
(PDF)

**S1 Fig. Network of frailty, quality of life and illness perception for participants recruited before COVID-19 pandemic.** Abbreviations: A/D, Anxiety/Depression; UA, Usual activities; P/D, Pain/Discomfort; MBLT, Mobility; COH, Coherence; CONS, Consequences; ER, Emotional response; IC, Illness concern; ID, Identity; PC, Personal control; TC, Treatment control; TIME, Timeline.
(TIF)

## Acknowledgments

We would like to thank all research participants for participating in our study.

## Author Contributions

**Conceptualization:** Anargyros Kapetanakis, Dimitrios Kyrou, Iliana Ntourou, Nikolaos Vrontaras, Ioannis Katsarolis, Christina Karamanidou.

**Data curation:** Anargyros Kapetanakis, Georgios Karakatsoulis, Dimitrios Kyrou.

**Formal analysis:** Anargyros Kapetanakis, Georgios Karakatsoulis.

**Investigation:** Olga Tsachouridou, Maria Meliou, Dimitrios Basoulis, Konstantinos Protopapas, Vasilis Petrakis, Leonidia Leonidou, Simeon Metallidis, Maria Chini, Mina Psichogiou, Anastasia Antoniadou, Periklis Panagopoulos, Charalambos Gogos.

**Methodology:** Anargyros Kapetanakis, Dimitrios Kyrou, Iliana Ntourou, Nikolaos Vrontaras, Christina Karamanidou.

**Project administration:** Anargyros Kapetanakis, Christina Karamanidou.

**Resources:** Olga Tsachouridou, Maria Meliou, Dimitrios Basoulis, Konstantinos Protopapas, Vasilis Petrakis, Leonidia Leonidou, Simeon Metallidis, Maria Chini, Mina Psichogiou, Anastasia Antoniadou, Periklis Panagopoulos, Charalambos Gogos.

**Supervision:** Christina Karamanidou.

**Validation:** Anargyros Kapetanakis, Georgios Karakatsoulis, Dimitrios Kyrou, Iliana Ntourou, Nikolaos Vrontaras, Olga Tsachouridou, Maria Meliou, Dimitrios Basoulis, Konstantinos Protopapas, Vasilis Petrakis, Leonidia Leonidou, Simeon Metallidis, Maria Chini, Mina

Psichogiou, Anastasia Antoniadou, Periklis Panagopoulos, Charalambos Gogos, Christina Karamanidou.

**Visualization:** Anargyros Kapetanakis, Christina Karamanidou.

**Writing – original draft:** Anargyros Kapetanakis, Georgios Karakatsoulis, Dimitrios Kyrou, Iliana Ntourou, Christina Karamanidou.

**Writing – review & editing:** Anargyros Kapetanakis, Georgios Karakatsoulis, Dimitrios Kyrou, Iliana Ntourou, Nikolaos Vrontaras, Olga Tsachouridou, Maria Meliou, Dimitrios Basoulis, Konstantinos Protopapas, Vasilis Petrakis, Leonidia Leonidou, Ioannis Katsarolis, Simeon Metallidis, Maria Chini, Mina Psichogiou, Anastasia Antoniadou, Periklis Panagopoulos, Charalambos Gogos, Christina Karamanidou.

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
