## [Decision Letter · Decision Letter 0]

8 Aug 2023

PONE-D-23-13247The impact of frailty and illness perceptions on quality of life among people living with HIV in Greece: a network analysisPLOS ONE

Dear Dr. Kapetanakis,

Thank you for submitting your manuscript to PLOS ONE. After careful consideration, we feel that it has merit but does not fully meet PLOS ONE’s publication criteria as it currently stands. Therefore, we invite you to submit a revised version of the manuscript that addresses the points raised during the review process.

We look forward to receiving your revised manuscript.

Kind regards,

Mario Ulises Pérez-Zepeda, M.D., Ph.D.

Academic Editor

PLOS ONE

Journal Requirements:

“This study is a collaborative research project that is supported and funded by Gilead Sciences Hellas (Medical Affairs).”

“I have read the journal's policy and the authors of this manuscript have the following competing interests: IK is an employee of Gilead Sciences Hellas and Cyprus (Medical Affairs). CK has received a grant from Gilead Sciences Hellas, paid to her institution, to support this study.”

We note that one or more of the authors are employed by a commercial company: Gilead Sciences Hellas and Cyprus

Reviewers' comments:

Reviewer's Responses to Questions

**Comments to the Author**

1. Is the manuscript technically sound, and do the data support the conclusions?

Reviewer #1: No

Reviewer #2: Yes

2. Has the statistical analysis been performed appropriately and rigorously? 

Reviewer #1: I Don't Know

Reviewer #2: I Don't Know

3. Have the authors made all data underlying the findings in their manuscript fully available?

Reviewer #1: No

Reviewer #2: No

4. Is the manuscript presented in an intelligible fashion and written in standard English?

Reviewer #1: Yes

Reviewer #2: Yes

5. Review Comments to the Author

Reviewer #1: Overall impression is very good. Well-written English, even I might say excellent. The paper raises an important topic.

Unfortunately, it lacked clarification of results; meaning, the network diagrams presented. Demographic and clinical data are supposed to be presented in a table, not in S1; additionally, not readable. It is not enough to state, “slow walking” (what are the values) etc. Methods are lacking what tools were used to put certain patient in certain frailty category. Also, it is not mentioned what comorbidities patients have.

In general, in my opinion the paper lacked ground representation of patients/participants and their assessment; which leaded authors to the conclusions that they draw. In Limitations section I think it is unnecessary to write limits after letters a)etc.

Line 55 „U=U” term should be explained

Line 58 QoL term should be explained first and then used as abbreviation.

Line102 in my opinion ; before and

Line 108 maybe only participants

Line 124 measurements not measures, even better wording would be methods

Line 180 package, package repetition

Line 186 444 males

Lines 186-187 age number±SD value

Line 188 table in S1 table? Repetition

Line 223 no footnote under the table

Line 232 what for the clarification “bold values…”

Reviewer #2: This paper presents relevant data in a logic and intelligible fashion. The topic has gained attention and the findings are in line with what has been previously reported. A major setback is the lack of access to the data for verification of the analyses. It should be possible to anonymize the data for public access. Another relevant piece of information that is missing is how the COVID-19 pandemic affected the data collection and might have affected the participants' answers to nearly all of the instruments that were applied, given the fact that de period of data collection coincided with the first wave of the pandemic.

6. PLOS authors have the option to publish the peer review history of their article (what does this mean?). If published, this will include your full peer review and any attached files.

Reviewer #1: No

Reviewer #2: **Yes: **Raul Hernan Medina Campos

---

## [Author Response · Author response to Decision Letter 0]

22 Sep 2023

Reviewer #1:

• Overall impression is very good. Well-written English, even I might say excellent. The paper raises an important topic.

Reply: Thank you for your constructive review of our work. It was very helpful to 

revise our manuscript.

• Unfortunately, it lacked clarification of results; meaning, the network diagrams presented.

Reply: In lines 263-313 we present and explain the results regarding network analysis. Additionally, in statistical analysis (lines 169-184) we present how we performed network analysis.

• Demographic and clinical data are supposed to be presented in a table, not in S1; additionally, not readable.

Reply: Thank you for this comment. Although it is common practice in papers with network analysis to skip the detailed presentation of basic characteristics, especially, if they are already published, we can understand from your comment that it makes the paper less readable. So, we chose to include the table in the main manuscript. We hope that it is more readable now.

• It is not enough to state, “slow walking” (what are the values) etc. Methods are lacking what tools were used to put certain patient in certain frailty category. 

Reply: For the frailty assessment we used frailty criteria as described by Fried et al. More specifically, we have provided a brief description of used frailty criteria with values and specific cut-offs for frailty in supplementary data.

• Also, it is not mentioned what comorbidities patients have.

Reply: Although comorbidities are not the focal point of this paper, we acknowledge the need of this information, so we included them in the first table with the demographic and clinical data.

• In general, in my opinion the paper lacked ground representation of patients/participants and their assessment; which leaded authors to the conclusions that they draw.

Reply: This is a secondary analysis of the data collected during the HIV Holistic Assessment program in Greece. The ground presentation of participants and their assessment are presented in our first publication as it is mentioned in the methodology. We hope that now after the inclusion of the supplementary table in the main manuscript, the addition of comorbidities and our reply in a previous comment regarding frailty assessment, the relevant information is fully provided.

• In the Limitations section I think it is unnecessary to write limits after letters a) etc.

Reply: Done.

• Line 55 „U=U” term should be explained 

Reply: Done.

• Line 58 QoL term should be explained first and then used as abbreviation.

Reply: QoL term was explained first in line 57.

• Line102 in my opinion ; before and

Reply: We cannot find the specific point. Please clarify this point.

• Line 108 maybe only participants

Reply: We mentioned “people living with HIV” in order to point out the possible generalization of our findings.

• Line 124 measurements not measures, even better wording would be methods

Reply: Done.

• Line 180 package, package repetition

Reply: Done.

• Line 186 444 males

Reply: Done.

• Lines 186-187 age number±SD value

Reply: Done.

• Line 188 table in S1 table? Repetition

Reply: S1 table was the name of the supplementary file. As we have added the supplementary table in the main manuscript, this comment is not applicable anymore.

• Line 223 no footnote under the table

Reply: We have added footnote under the table.

• Line 232 what for the clarification “bold values…”

Reply: The bold values represent all p-values <0.05. We have added a footnote about it.

Reviewer #2:

• This paper presents relevant data in a logic and intelligible fashion. The topic has gained attention and the findings are in line with what has been previously reported.

Reply: Thank you for your constructive review of our work. It was very helpful to revise our manuscript.

• A major setback is the lack of access to the data for verification of the analyses. It should be possible to anonymize the data for public access.

Reply: Due to the sensitive nature of the patient data contained within, our preference is to make the database available exclusively to researchers who reach out to us, rather than granting public access, as it raises concerns about the privacy of the patients involved.

• Another relevant piece of information that is missing is how the COVID-19 pandemic affected the data collection and might have affected the participants' answers to nearly all of the instruments that were applied, given the fact that the period of data collection coincided with the first wave of the pandemic.

Reply: Thank you for your comment. Indeed, some of our participants were recruited during the first wave of the pandemic. As the study design did not take into consideration this condition, we cannot reach safe conclusions regarding the role and impact of the pandemic in the data collection. We acknowledged the possible sampling bias and the COVID-19 pandemic’s influence on QoL in limitations. For this reason, we performed a sub-network analysis including only the participants recruited before the first lockdown and we found similar results regarding the associations (supplementary material) indicating that our findings were not affected by COVID-19 pandemic.

---

## [Editor Report · Decision Letter 1]

28 Sep 2023

The impact of frailty and illness perceptions on quality of life among people living with HIV in Greece: a network analysis

PONE-D-23-13247R1

Dear Dr. Kapetanakis,

We’re pleased to inform you that your manuscript has been judged scientifically suitable for publication and will be formally accepted for publication once it meets all outstanding technical requirements.

Kind regards,

Mario Ulises Pérez-Zepeda, M.D., Ph.D.

Academic Editor

PLOS ONE
---

## [Editor Report · Acceptance letter]

10 Nov 2023

PONE-D-23-13247R1 

The impact of frailty and illness perceptions on quality of life among people living with HIV in Greece: a network analysis 

Dear Dr. Kapetanakis:

I'm pleased to inform you that your manuscript has been deemed suitable for publication in PLOS ONE. Congratulations! Your manuscript is now with our production department. 

Kind regards, 

on behalf of

Dr. Mario Ulises Pérez-Zepeda 

Academic Editor

PLOS ONE